# Hybrid supercapacitors for reversible control of magnetism

Alan Molinari[1], Philipp M. Leufke[1], Christian Reitz[1], Subho Dasgupta[1,†], Ralf Witte[1], Robert Kruk[1] & Horst Hahn[1,2]

Electric field tuning of magnetism is one of the most intensely pursued research topics of recent times aiming at the development of new-generation low-power spintronics and microelectronics. However, a reversible magnetoelectric effect with an on/off ratio suitable for easy and precise device operation is yet to be achieved. Here we propose a novel route to robustly tune magnetism via the charging/discharging processes of hybrid supercapacitors, which involve electrostatic (electric-double-layer capacitance) and electrochemical (pseudocapacitance) doping. We use both charging mechanisms—occurring at the $La_{0.74}Sr_{0.26}MnO_3$/ionic liquid interface to control the balance between ferromagnetic and non-ferromagnetic phases of $La_{1-x}Sr_xMnO_3$ to an unprecedented extent. A magnetic modulation of up to $\approx 33\%$ is reached above room temperature when applying an external potential of only about 2.0 V. Our case study intends to draw attention to new, reversible physico-chemical phenomena in the rather unexplored area of magnetoelectric super-capacitors.

[1] Institute of Nanotechnology (INT), Karlsruhe Institute of Technology (KIT), Hermann-von-Helmholtz-Platz 1, 76344 Eggenstein-Leopoldshafen, Germany. [2] KIT-TUD-Joint Research Laboratory Nanomaterials, Technical University Darmstadt, Jovanka-Bontschits-Strasse 2, 64287 Darmstadt, Germany. † Present address: Department of Materials Engineering, Indian Institute of Science (IISc), Bangalore 560012, India. Correspondence and requests for materials should be addressed to A.M. (email: alan.molinari@kit.edu) or to H.H. (email: horst.hahn@kit.edu).

In quest of more energetically effective data storage and processing, sensing and actuation, the idea of using an electric field to control magnetism has been investigated in various fascinating ways[1–5]. A common and direct approach to realize the magnetoelectric (ME) effect is via manipulation of the electronic structure of a magnetic material by voltage-induced charge doping[6]. Typically, two possible charging mechanisms may occur at solid/solid or solid/liquid interfaces. The first involves non-faradaic electrostatic doping, where the charge carriers are electrostatically separated at the interface between a magnetic material and a ferroelectric[7–10], a dielectric[11] or an electrolyte[12,13], in analogy to a parallel plate capacitor. The second implies faradaic electrochemical doping, where redox reactions with exchange of charge carriers[14–17] occur across the interface between a magnetic material and a different chemical species, resembling the behaviour of an electrochemical cell.

It has been experimentally shown that the strength of the ME effect at the interface is directly related to the electrical capacitance[6,9]. In this regard, supercapacitors (SCs), so far primarily utilized for energy storage and delivery, provide attractive tools to reversibly and robustly control magnetism. They can accumulate charge electrostatically (as in electric-double-layer capacitors[18,19]), electrochemically (as in metal-oxide pseudocapacitors[20–23]) or by a combination of both (as in hybrid SCs[24,25]). In particular, pseudocapacitors can reach values of capacitance[23] as high as $\sim 10^3 \, \mu F \, cm^{-2}$, which allows for surface charge densities appreciably beyond ferroelectrics (for example, $\sim 150 \, \mu C \, cm^{-2}$ for (ref. 26) $BiFeO_3$). Further, they feature an exceptional degree of reversibility on charge/discharge cycling[27,28].

To date, intensive efforts have been aimed at improving the SCs performance in terms of energy storage capacity and delivery, but the idea of utilizing a pseudocapacitor or a hybrid SC, per se, to control magnetism has not been embraced yet. In view of the recent discovery[29] of pseudocapacitive reactions in $LaMnO_3$, the class of magnetic manganese-based perovskite oxides naturally qualifies as a system of interest for modulation of magnetic properties via both electrostatic and electrochemical stimuli. In particular, $La_{1-x}Sr_xMnO_3$ (LSMO), owing to its high Curie temperature, can certainly be considered as a technologically relevant prototype system. This strongly correlated oxide features a rich magnetoelectronic phase diagram[30] encompassing a variety of magnetic states[31] that are determined by the charge state of the manganese ions ($Mn^{3+/4+}$).

In this work we demonstrate that strong and reversible control of magnetism can be realized in ME hybrid SCs. Here by means of in situ synchronized superconducting quantum interference device (SQUID) magnetometry and cyclic voltammetry (CV), we track the magnetic response of a LSMO single-crystal thin film while modulating its surface charge density using an ionic liquid. We prove that the charging/discharging processes at the solid/liquid interface are those of a hybrid SC where electrostatic doping (electric-double-layer capacitance, $C \approx 10 \, \mu F \, cm^{-2}$) is gradually replaced by faradaic electrochemical reactions (pseudocapacitance, $C$ up to $\approx 180 \, \mu F \, cm^{-2}$) on progressive increase of the external voltage. We make use of both charging mechanisms to tune the magnetic interactions in LSMO by manipulating the oxidation state of the Mn ions. We exploit surface charge densities as high as $\approx 270 \, \mu C \, cm^{-2}$ to comprehensively explore the surface magnetoelectronic phase diagram of LSMO.

## Results

**Capacitive and pseudocapacitive charging regimes.** The experiments were performed on atomically smooth $La_{0.74}Sr_{0.26}MnO_3$ thin films ($\approx 13 \, nm$), epitaxially grown on (001)-oriented $SrTiO_3$ substrates (see Methods section).

An electrochemical tuning cell (see schematic in Fig. 1a), composed of a LSMO film (working electrode), a high-surface-area carbon cloth (counter electrode) separated by an insulating porous glass fibre soaked in ionic liquid (diethylmethyl

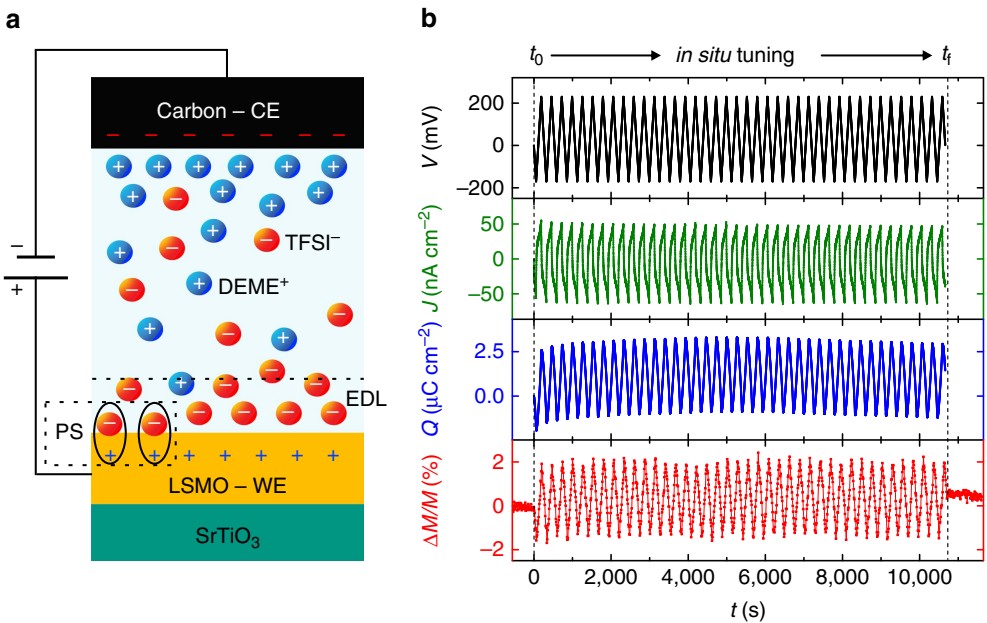

**Figure 1 | Sketch of the device and in situ measurement principle.** (**a**) Schematic of the electrochemical tuning cell: a $La_{0.74}Sr_{0.26}MnO_3$ (LSMO) single-crystal thin film ($\approx 13 \, nm$) grown on a $SrTiO_3$ substrate and a high-surface-area carbon cloth serve as working (WE) and counter (CE) electrodes, respectively. The electrodes are separated by an insulating glass fibre (not shown). On application of an external voltage, the ions of the ionic liquid ($DEME^+$-$TFSI^-$) form physical or chemical bonds with the LSMO surface leading to electrostatic (electric-double-layer (EDL) capacitance) or electrochemical (pseudocapacitance (PS)) charge carrier doping, respectively. Both mechanisms allow for manipulating the magnetic state of LSMO. (**b**) Example of in situ tuning experiment performed at 323 K: the magnetic response (in red) reversibly follows the surface charge modulation (in blue), calculated by integrating the measured current density (in green), on repetitive cycling of the external potential (in black).

(2-methoxyethyl)ammonium bis(trifluoromethylsulfonyl)imid, DEME-TFSI), was inserted into a SQUID magnetometer and connected to a potentiostat. The set-up allowed for quantitative and quasi-continuous *in situ* measurements of the magnetization response concurrently to modulation of the interface charge density. During the CV measurements, while the external potential $V(t)$ was being ramped, the current density $J(t)$ flowing towards the LSMO electrode was monitored, and by integrating it, the surface charge $Q(t)$ at the LSMO/ionic liquid interface was calculated. Simultaneously, the magnetic response $M(t)$ was measured. An illustrative example of a typical tuning experiment performed at 323 K, revealing the high degree of reversibility achieved above room temperature, is provided in Fig. 1b. In the initial state (before $t_0$), the magnetization is constant. Afterwards, during CV cycling (from $t_0$ to $t_f$), the magnetic signal follows the surface charge modulation in a highly reproducible zigzag-like fashion with low-noise fluctuations. After $t_f$, the magnetization recovers its initial value.

The figure of merit in CV experiments is the capacitance, defined as $C = \Delta Q/\Delta V_{Qmax}$, where $\Delta Q$ is the difference between positive and negative maximum surface charge and $\Delta V_{Qmax}$ is the difference in the respective voltages. High values of capacitance are certainly desirable for tuning the magnetization because they directly result in large modulations of interface charge density $\Delta Q$. Thus, the higher the charge carrier accumulation at the LSMO electrode is, the larger the number of Mn ions to undergo a change in oxidation state will be. This, in turn, enables a wider portion of the LSMO magnetoelectronic phase diagram to be probed in a full voltage sweep (see below for in-depth discussion).

Owing to a number of very sensitive control parameters, such as temperature, applied voltage and electrode/electrolyte chemical compatibility, a clear determination of whether the capacitance originates from electrostatic, electrochemical charging or a combination of both, is a non-trivial task[32–34]. Here, to reliably discern between faradaic and non-faradaic charging regimes, a series of systematic CV measurements was conducted for different temperatures, potential windows and voltage sweep rates.

Initially, a temperature-dependence survey was carried out from 330 down to 220 K, that is, slightly above the LSMO Curie point down to the DEME-TFSI glass transition[35]. The surface charge modulation was intentionally kept at the small value of $\Delta Q \approx 4\,\mu C\,cm^{-2}$ using a working potential window of $\Delta V = V^{max} - V^{min} \approx 400\,mV$. Thus, electrochemical reactions at the LSMO/ionic liquid interface were hindered. Throughout the whole investigated temperature range, the $J(V)$ characteristics (see Supplementary Fig. 2) featured the behaviour of an ideal capacitor with symmetric and rectangular shape of the charging/discharging processes and a remarkable reversibility on cycling. The calculated capacitance $C \approx 10\,\mu F\,cm^{-2}$ was virtually constant (see Supplementary Table 1) and consistent with the expected double-layer capacitance values reported for metallic conductive electrodes in ionic liquids[36–39]. These findings give a strong indication that under application of small external voltages, the CV measurements were performed within a regime dominated by electrostatic doping.

Afterwards, an isothermal charge-dependence study (Fig. 2a), with progressive expansion of the potential window $\Delta V$ from 0.2 up to 3.7 V, was carried out at the lowest temperature of 220 K. Interestingly, up to $\sim 2.4\,V$, the $J(V)$ characteristics still exhibited a high level of reversibility with a nearly perfect rectangular shape. However, a systematic increase in current density, and therefore capacitance, was observed. These features represent the fingerprint of ideal pseudocapacitance[21,22,40], denoting that electrode charging/discharging is driven also by interfacial electrochemical reactions. At a certain limit, when $\Delta V$ was considerably expanded

up to 3.7 V, the $J(V)$ curves displayed more pronounced electrochemical features.

The trend of the calculated capacitance, which increases from $\approx 10$ up to $\approx 85\,\mu F\,cm^{-2}$, is illustrated in Fig. 2b. The onset of pseudocapacitance seems to already develop at low voltages since the double-layer capacitance of standard metallic electrodes exhibits exactly the opposite behaviour[39]. Beyond a threshold value of $\Delta V \sim 2.4\,V$, the steeper increase in $C$ heralds a transition to a pseudocapacitive-dominated regime. Notably, at the biggest applied potential window of 3.7 V, the surface charge modulation reached a value of $\approx 270\,\mu C\,cm^{-2}$.

It is known[21,22,29] that pseudocapacitance involves redox reactions with faradaic charge transfer occurring at the surface and/or in the bulk of the electrode. Here ionic intercalation into LSMO is unlikely as the DEME[41] and TFSI[37] ions are nearly twice the size of the LSMO unit cell[30]. A minor contribution due to oxygen ions migration into LSMO may be attributed to the decomposition of residual traces of water[42] present in the ionic liquid (see Methods section). This is compatible with the appearance of small irreversible redox peaks on expansion of the potential window up to 3.7 V (see arrows in Fig. 2a). The interpretation that surface pseudocapacitance plays a major role in the electrochemical doping of LSMO is corroborated by a voltage ramp rate study performed at the fixed potential windows of 0.4 and 2.9 V (Fig. 2c). On the one hand, the linear behaviour of the current density indicates the presence of fast surface charging/discharging processes[20,43], resembling the charge-transfer kinetics of a capacitor rather than of an electrochemical cell. On the other hand, the calculated slopes confirm the radical increase in capacitance on expansion of the potential window.

These findings imply that the LSMO/ionic liquid system behaves as an archetypal hybrid SC. A plausible scenario is that the electric field-driven attraction of DEME and TFSI ions to the LSMO surface changes from physisorption to chemisorption when the potential window is increased. Consequently, the charging processes move from electrostatic (electric-double-layer capacitance) to electrochemical (surface redox pseudocapacitance) doping.

The behaviour of $C$ on isothermal increase of the applied voltage was also studied at higher temperatures (see Supplementary Note 4 and Supplementary Fig. 5), where the thermally activated nature of electrochemical reactions[16] precipitated the occurrence of pseudocapacitance. A maximum value of $C \approx 180\,\mu F\,cm^{-2}$ was achieved at around 320 K under the application of 1.8 V, leading to a surface charge density of $\Delta Q \approx 200\,\mu C\,cm^{-2}$.

**Electric field control of magnetism**. In the following, the extensive and flexible control of the LSMO magnetic properties by means of both electrostatic and pseudocapacitive charging will be described.

A magnetic field-cooled measurement performed before *in situ* tuning experiments (Fig. 3a) revealed a Curie temperature of $T_C \approx 323\,K$. Afterwards, the effect of a nearly constant surface charge modulation $\Delta Q \approx 4\,\mu C\,cm^{-2}$ on the LSMO magnetic response was investigated from above $T_C$ down to 220 K. The peak-to-peak variation in magnetization was defined as $\Delta M = M_{acc}^{h^+} - M_{depl}^{h^+}$, that is, the difference between the maximum magnetization in hole accumulation and depletion states.

Figure 3b represents the behaviour of $\Delta M_\sigma$, determined by normalizing $\Delta M$ with respect to the LSMO surface unit cell area (u.c.²). The definition accounts for the interfacial nature of the tuning effect by assuming in first approximation a Thomas-Fermi screening length[7,9] for the electric field of $\sim 0.2–0.4\,nm$. The trend of $\Delta M/M$ that gives the overall tuning effect compared to the untuned LSMO magnetization is depicted in Fig. 3c.

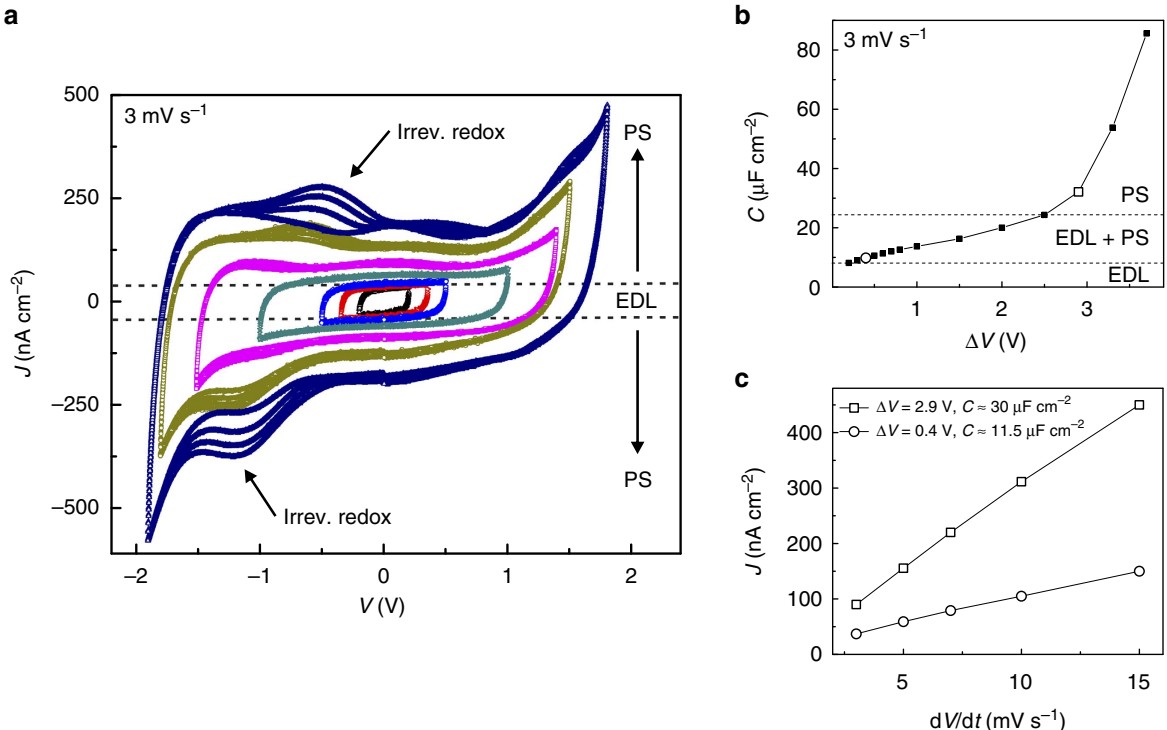

**Figure 2 | Transition from capacitive to pseudocapacitive charging at 220 K.** (**a**) Current density—voltage characteristics upon expansion of the potential window ($0.2\,V < \Delta V < 3.7\,V$) using a scan rate of $3\,mV\,s^{-1}$. Each curve corresponds to four consecutive CV cycles. The dashed lines qualitatively delimit the operative region expected for an ideal EDL capacitor. The increment in current density on increasing the applied $\Delta V$ indicates a progressive PS behaviour. The arrows indicate the points where irreversible redox reactions start to appear. (**b**) Calculated capacitance versus applied voltage for a scan rate of $3\,mV\,s^{-1}$. The trend of $C$ indicates a gradual transition from EDL capacitance to PS up to $\Delta V \sim 2.4\,V$. Beyond this value, the steep increase in $C$ denotes the predominance of PS. (**c**) Voltage ramp rate dependence of the current density for a particular potential window of 0.4 and 2.9 V. In **b**,**c**, the experimental errors are within a 5% accuracy.

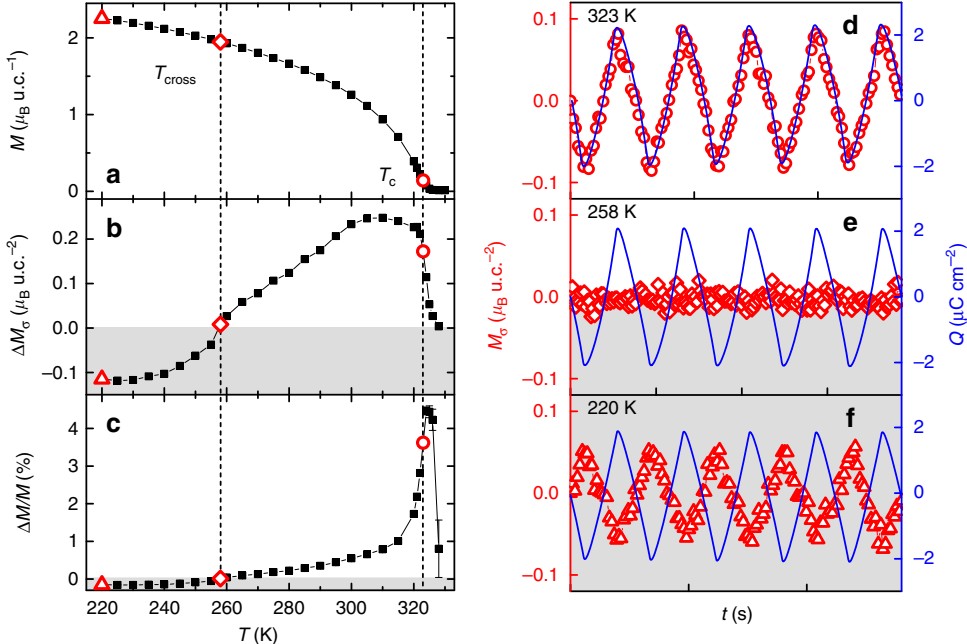

**Figure 3 | Temperature-dependence study on constant surface charge modulation $\Delta Q \approx 4\,\mu C\,cm^{-2}$.** (**a**) Magnetic field-cooled curve performed before *in situ* tuning revealing a $T_C \approx 323\,K$. (**b**,**c**) Response of the magnetic modulation normalized per LSMO surface unit cell (**b**) and with respect to the volume magnetization (**c**). A crossover temperature $T_{cross} \approx 258\,K$ separates in-phase (white area) and anti-phase (grey area) magnetic responses with respect to the surface charge modulation. (**d**–**f**) Time-resolved magnetic response at $T_C$ (**d**), $T_{cross}$ (**e**) and 220 K (**f**). Each time step in **d**–**f** on the abscissa corresponds to 500 s.

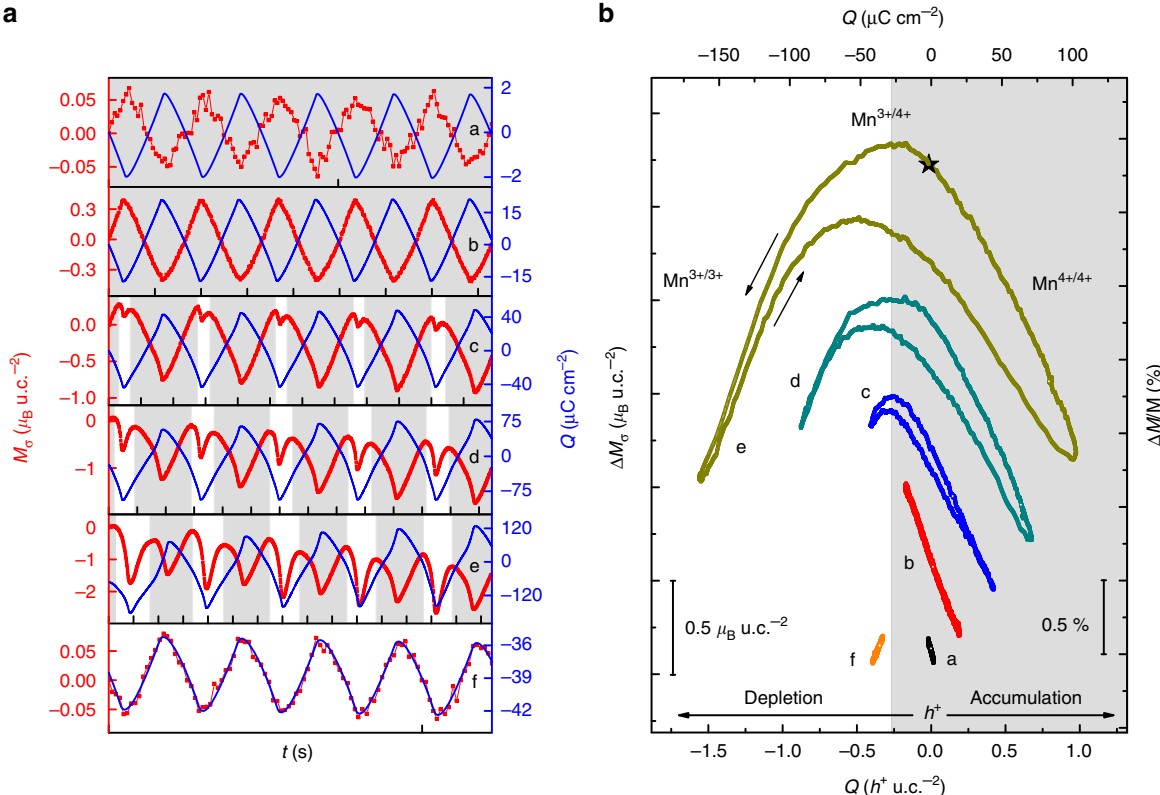

**Figure 4 | Isothermal charge-dependence study at 220 K. (a)** Time-resolved magnetic responses on sequential increase of the surface charge modulation $\Delta Q$ from $\approx 1.5$ up to $\approx 270\,\mu C\,cm^{-2}$. The expected anti-phase response (plots a,b) reveals an in-phase splitting of the $M(t)$ curve (plots c, d and e) on progressive electron doping, like the behaviour observed above $T_{cross}$ for a small potential window. The application of a bias voltage $V_b \approx -1.3\,V$ yielded an in-phase response (plot f). Each time step on the abscissa corresponds to 800 s. **(b)** Behaviour of the magnetization as a function of the surface charge doping. The slope of the $M(Q)$ curves corresponds to the ME coupling coefficients $\alpha$. The small variation in $\Delta M/M$ underlines the interfacial nature of the magnetic tuning effect at low temperature. In **a,b**, the white and grey areas qualitatively separate the data with the LSMO magnetization responding to the applied surface charge as above or below $T_{cross}$, respectively.

The study of the temperature dependence revealed that above $T_C$ the magnetic response $M(t)$ was negligible while sweeping the surface charge $Q(t)$. On reaching $T_C$, a steep tuning effect with $M(t)$ changing in-phase with $Q(t)$ (Fig. 3d) was observed. After reaching a maximum magnetic modulation of $\approx 4.5\%$, a further decrease in temperature brought about a gradual reduction of the magnetic signal until it virtually vanished at $T_{cross} \approx 258\,K$ (Fig. 3e). Below this threshold, a crossover with a reverse in sign of the magnetic modulation was observed, that is, $M(t)$ changed to anti-phase with respect to $Q(t)$ (Fig. 3f). The existence of a crossover point has already been shown in other LSMO-tuning studies[7,9,12,44], but here $T_{cross}$ reaches the highest temperature reported.

A broader inspection of the surface magnetoelectronic phase diagram of LSMO started with tracking down the magnetic response during isothermal progressive expansion of the potential window (0.2 V < $\Delta V$ < 3.7 V). It turned out that the high value of $T_{cross}$ together with the achievable surface charge modulation of up to $\Delta Q \approx 270\,\mu C\,cm^{-2}$ had a deep influence on the magnetism.

Focusing on the results obtained at 220 K, from small applied voltages up to $\sim 2.9\,V$, the magnetic signal was in anti-phase with respect to the surface charge modulation (plots a,b in Fig. 4a), in agreement with the results of temperature dependence. One of the main findings is that above this threshold voltage, the $M(t)$ curve manifested a splitting on the electron accumulation side (plot c in Fig. 4a) with the distinctive in-phase characteristic observed above $T_{cross}$ for a small potential window. As the potential window was further increased (plots d,e in Fig. 4a), this

trend became more and more pronounced. This phenomenon allows for controlling the sign of the tuning effect under the application of a bias voltage. Indeed, by setting a negative bias $V_b \approx -1.3\,V$, the magnetic signal became immediately in-phase with the charge modulation (plot f in Fig. 4a). The versatility of LSMO magnetic responses achieved by simply adjusting the external voltage demonstrates an unprecedented control of the interfacial ME coupling.

To quantify the strength of the ME effect we define the coupling coefficient $\alpha = \Delta M/\Delta Q$. Its introduction as a figure of merit directly stems from the access to reliable, quantitative data on surface charge density and magnetization as given by the potentiostat and the SQUID magnetometer. The calculation of the slope of the $M(Q)$ curves allowed for precise determination of $\alpha$ for both temperature-dependence and isothermal charge-dependence surveys. Regarding the former, as a direct consequence of the trend of the $\Delta M(T)$ curves, the sign of $\alpha$ was positive or negative above or below $T_{cross}$, respectively, with values ranging between $+6.4\,\mu_B/h^+$ and $-3.2\,\mu_B/h^+$ (see also Supplementary Note 2 and Supplementary Fig. 3). The latter study (Fig. 4b) revealed that on increasing $\Delta Q$ at a fixed temperature, the system behaved as being toggled back and forth across $T_{cross}$. Indeed, at 220 K it was possible to reversibly set the ME coupling coefficient to positive ($\alpha \sim +2.2\,\mu_B/h^+$) or negative ($\alpha \sim -3\,\mu_B/h^+$) values with respect to a plateau region where a maximum magnetization was reached (see also Supplementary Table 2). Furthermore, it is worth underlying that the magnetic tuning presented $\Delta M/M$ ratios of only a few per

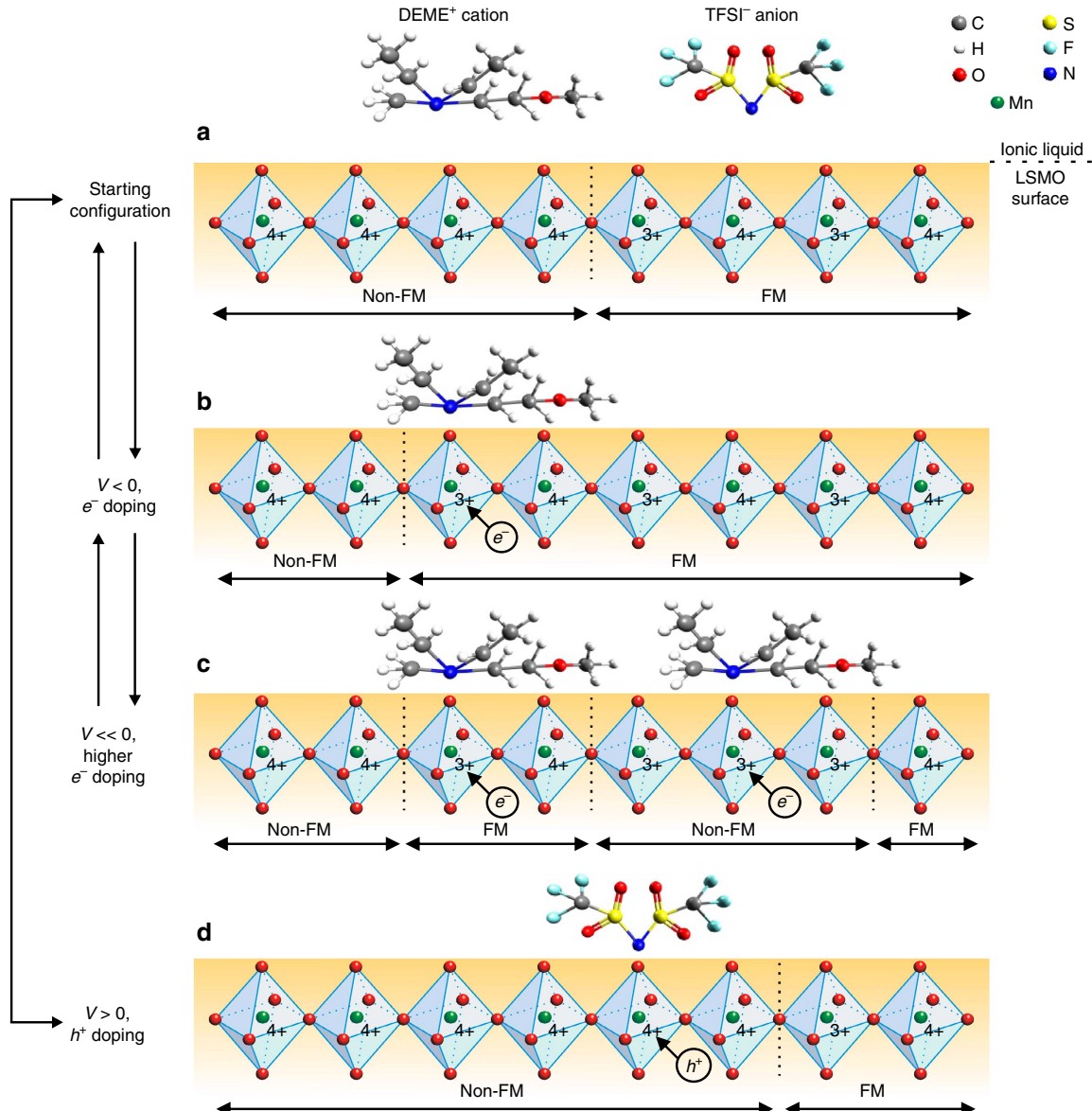

**Figure 5 | Mechanism of electric field tuning of magnetoelectronic phase separation at the LSMO/ionic liquid interface.** For the sake of clarity, not the entire perovskite unit cell of LSMO, but only the oxygen octahedron containing one Mn ion is shown. The sketches should be read together with the trend of the $M(Q)$ curves in Fig. 4b. (**a**) Initial surface magnetoelectronic configuration of LSMO with two distinguished FM and non-FM domains. (**b**) Expansion of the FM domain on electron doping (optimization of $Mn^{3+/4+}$ balance). (**c**) Shrinkage of the FM domain on further electron doping (increase in $Mn^{3+/3+}$ balance). (**d**) Shrinkage of the FM domain on hole doping (increase in $Mn^{4+/4+}$ balance).

cent in all the measurements performed well below $T_C$, corroborating the idea that surface effects are predominantly involved, as already concluded. Bigger magnitudes in magnetic modulation were encountered only in proximity of $T_C$ (see later discussion).

A tentative approach[7,9,44] to interpret the interfacial ME phenomena starts with the LSMO bulk phase diagram[30]. Considering the doping region around $x \approx 0.26$, each introduced hole (electron), regardless of whether through electrochemical or electrostatic doping, results in a shift of $T_C$ towards higher (lower) temperatures. Concurrently (and with the opposite trend), the ground state magnetization ($M_{Mn} = (4 - x)\mu_B$ u.c.$^{-1}$) decreases (increases) on hole (electron) doping in accordance with a band filling model where each introduced positive (negative) carrier contributes with a magnetic moment of $-(+)1\ \mu_B$. The competition between these two effects, with the former prevailing in proximity of $T_C$ and the latter at low temperatures, is clearly

evidenced in the temperature-dependence survey by the presence of a crossover point $T_{cross}$, which separates the in-phase and anti-phase regimes of $M(t)$ over $Q(t)$. Further details are discussed in Supplementary Note 3 and Supplementary Fig. 4.

Although this phenomenological picture is sufficient to qualitatively describe the observed general trends, it notably fails[7,9,44] when quantitative data are taken into account. To capture the essence of a plausible microscopic picture, one should bear in mind the following considerations based on the experimental findings at 220 K (see also Supplementary Table 2), which are unaffected by the possible critical effects occurring close to $T_C$.

First, the interfacial nature of the magnetic tuning effect together with charge density modulations of up to $\Delta Q \approx 270\ \mu C\ cm^{-2}$ (corresponding to about $\pm 1 h^+$ u.c.$^{-2}$) imply that the LSMO surface magnetoelectronic configuration can be virtually probed across the entire range of possible hole-doped states of

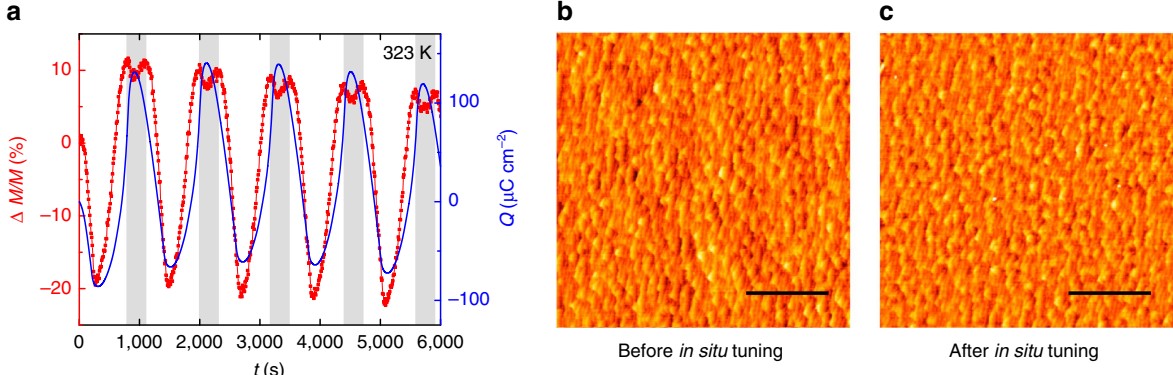

**Figure 6 | Tuning of magnetism in proximity of $T_C$ and LSMO surface characterization.** (**a**) Time-resolved magnetic response featuring a $\approx 33\%$ tuning effect on modulation of the surface charge density ($\Delta Q \approx 200 \,\mu C \, cm^{-2}$). The expected in-phase response (white area) of $M(t)$ with respect to $Q(t)$ reveals an anti-phase splitting (grey area) on the hole doping branch, akin to the behaviour encountered below $T_{cross}$ for a small potential window. (**b,c**) Atomic force micrographs of the LSMO surface before (**b**) and after (**c**) in situ tuning. In both **b,c**, scale bar, total height scale root-mean-square roughness are about 1 microm, 1.5 nm and 0.2 nm, respectively.

the phase diagram. It is known[31] that the interaction between adjacent $Mn^{3+/4+}$, $Mn^{3+/3+}$ and $Mn^{4+/4+}$ ions promotes ferromagnetic (FM), A-type and G-type antiferromagnetic (AF) ground states, respectively. Furthermore, at non-zero temperatures, paramagnetic (PM) states come into play as well.

Second, the absolute value of the ME coupling coefficient is $|\alpha| \sim 3 \,\mu_B/h^+$. This is not in line with the $1 \,\mu_B/h^+$ predicted by a band filling model, but rather points to the annihilation (creation) of an entire magnetic moment carried by one Mn ion ($M_{Mn} \approx 3.7 \,\mu_B$ u.c.$^{-1}$). Additionally, $|\alpha|$ is nearly constant over a broad range of investigated $\Delta Q$, already when pseudocapacitance contributes to surface charging. This suggests that electrostatic and electrochemical doping, in spite of the different physical and chemical origin, have a similar functional role in the magnetic phase control.

These experimental facts indicate that a microscopic model still coherent with the major phenomenological features is conceivable only by incorporating a magnetoelectronic phase separation phenomenon[45–47] at the LSMO surface. In a simplified scenario (see Fig. 5), the local electric field at the electrode/ionic liquid interface induces the accumulation of charge carriers at the LSMO terminating layers. The consequent change in oxidation state of the Mn ions promotes either the expansion or shrinkage of FM and non-FM (PM or AF) domains, at the expense of each other. Considering plot e in Fig. 4b, the initial increase in magnetization on electron doping can be explained as a gradual optimization of the $Mn^{3+/4+}$ ratio, hence fostering the growth of FM-coupled domains (Fig. 5b). During sweeping to higher electron concentrations, the oxidation state of the surface manganese ions progressively shifts towards a $Mn^{3+/3+}$ balance. Either AF or PM ordering is favoured (Fig. 5c), thus decreasing the overall magnetization. By reversing the voltage sign, the magnetization recovers up to a maximum point due to the restored $Mn^{3+/4+}$ interactions. By sweeping further along the hole accumulation side, the amount of $Mn^{4+/4+}$ domains increases (Fig. 5d). Owing to the expansion of non-FM states, a decrease in the total magnetization is again observed.

A series of analogous isothermal charge-dependence studies were also carried out at temperatures above $T_{cross}$. The results of the experiments performed at 270 K confirm the findings acquired at 220 K and corroborate the proposed microscopic model of electric field tuned magnetoelectronic phase separation (see Supplementary Note 4 and Supplementary Fig. 5).

The sensitivity of the system to surface charge modulation was also probed at the immediate proximity of $T_C$. A remarkable

overall magnetic modulation $\Delta M/M \approx 33\%$ was reached using a potential window of only $\Delta V \approx 1.8 \, V$ (Fig. 6a). The high-tuning effect can be explained by a deeper penetration depth of the electric field close to the insulator-metal transition, which corresponds[48] also to the para-ferromagnetic transition. Although the Thomas-Fermi screening length[7,9] in purely metallic manganites reported in the literature is typically of 0.2–0.4 nm, it has been demonstrated that around $T_C$ the electric field can effectively extend up to 5 nm in electrolyte-gated $La_{0.8}Ca_{0.2}MnO_3$ films[33]. Furthermore, in presence of magnetoelectronic phase separation a strongly inhomogeneous and much deeper penetration of the electric field is expected[47].

Eventually, after performing more than 1,000 CV cycles under different temperatures and applied voltages, atomic force microscopy was employed to analyse the surface morphology of LSMO. The striking similarity of the atomic force micrographs before and after in situ tuning (Fig. 6b,c), featuring in both cases an atomically smooth surface with one or half unit cell steps, demonstrates the astonishing non-destructive nature of electric-double-layer and pseudocapacitive charging.

## Discussion

Our results demonstrate that fully reversible and robust ME coupling can be realized in LSMO/ionic liquid hybrid SCs by exploiting surface electrostatic and electrochemical doping. From a fundamental scientific perspective, the interdisciplinary character of our study aims at bridging the fields of MEs and SCs. The quantitative data, such as precisely determined values of $\alpha$, extracted from the temperature and charge-dependence surveys provide a solid base for further theoretical modelling of the interfacial tuning phenomena. On the more practical side, the devices tested in a broad temperature range and at low voltages show an unprecedented reversibility and flexibility in controlling both the magnitude and sign of the magnetic response using electric-double-layer and surface pseudocapacitive charging. In general, we propose that surface pseudocapacitance should expand the portfolio of tools to control magnetism, which already comprises strain, electrostatic doping and bulk-intercalation chemistry. Our work, although focusing on LSMO manganites, intends to pave the way for a broader, in-depth research on other potential magnetically functional SCs, such as cobaltites, ferrites and nickelites. The huge and reversible charge densities attainable at low voltages via surface pseudocapacitance may lead to giant magnetic tuning effects with possible

applications in future, low-power electric field and spintronic-based microelectronics.

## Methods

**Synthesis and characterization of LSMO films.** $La_{0.74}Sr_{0.26}MnO_3$ thin films were grown via large-distance rf-magnetron sputtering at a pressure of 0.018 mbar and a deposition temperature of $\approx 650\,°C$ on single-crystal $SrTiO_3$ substrates[49]. Before *in situ* tuning, the films were comprehensively investigated with a variety of characterization techniques (see Supplementary Note 1 and Supplementary Fig. 1).

**Electrochemical tuning cell preparation.** An electrochemical cell composed of a LSMO film (surface area of $\approx 0.4\,cm^2$) as the working electrode, a high-surface-area carbon fibre cloth ($S_{BET} \approx 2,400\,m^2\,g^{-1}$, Whatman) as the counter electrode, a glass fibre filter disc (GF/A, Whatman) as the separator and an ionic liquid (DEME-TFSI 98.5%, Sigma-Aldrich) as the non-aqueous electrolyte, was assembled under glove box conditions (with $[O_2] < 1$ p.p.m. and $[H_2O] < 1$ p.p.m.). Before the cell preparation, the ionic liquid was purified from water contamination by keeping it under vacuum conditions ($\sim 10^{-3}$ mbar) at $120\,°C$ for 24 h; afterwards Karl Fischer titration revealed an $H_2O$ level $< 50$ p.p.m. Gold and kapton-coated copper wires were connected internally to working and counter electrodes, respectively, and externally to electrical feedthroughs which allowed the transfer of signals with a potentiostat ($\mu$Autolab Type3). Once prepared, the tuning cell was quickly transferred into a SQUID magnetometer (Quantum Design MPMS XL-5). The voltage and current outputs of the potentiostat were put in communication with two multimeters (Keithley 2001), which delivered the signals to the SQUID magnetometer. Finally, a software script allowed for real-time control and detection of magnetization, voltage and current signals.

***In situ* ME tuning measurements.** All the SQUID measurements were performed at an external magnetic field of 100 Oe applied parallel to the plane of the film; reversing the sign of the field resulted in a mirror image of the time-resolved magnetic modulation. Unless differently specified, all the CV measurements were carried out at a fixed voltage ramp rate of $3\,mV\,s^{-1}$. Owing to the two-electrode configuration of the tuning cell, all the working potential windows were set with respect to open circuit potential (OCP $\approx -100$ mV) as their starting point, and the voltage data were presented accordingly.

**Data availability.** All relevant data are available from the corresponding authors on reasonable request.

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

## Acknowledgements

We are grateful to R.A. Brand, J. Schmalian, B. Jeevanesan and M. Rosiejka for their interest and fruitful discussions. We thank A. Houari and A. Beck for the TEM and RBS characterization of the LSMO films, respectively. S.D. and H.H. thank the financial support from the DFG Grants DA 1781/1-1 and HA 1344/34-1, respectively. We acknowledge support by Deutsche Forschungsgemeinschaft and Open Access Publishing Fund of Karlsruhe Institute of Technology.

## Author contributions

H.H. and R.K. supervised the study. A.M. prepared and characterized the samples with the support of P.M.L. and R.W., respectively. A.M. performed the *in situ* tuning experiments with the help of R.K., C.R. and S.D. A.M. analysed the data with the support of R.K., S.D. and P.M.L. A.M. wrote the manuscript with assistance from R.K. and input from H.H., C.R., R.W. and S.D. All authors discussed the results and commented on the manuscript.

## Additional information

**Competing interests:** The authors declare no competing financial interests.

