## [Peer Review File · Nature Communications]

Reviewer #1 (Remarks to the Author):

In the current work, involving hybrid supercapacitance approach to control magnetism in LSMO is interesting, and the results presented are efficient compared to traditional way of using only EDL capacitance. The experimental results confirm the significant role of pseudo capacitance by creating large surface charge density in comparison with EDL capacitance as described in Figure 1 and 2. Further in Figure 3 and 4, surface charge modulation mechanism in LSMO is elaborated by studying $M(T)$ under constant charge modulation, and varying constant charge modulation under isothermal condition to insight on mechanism of magnetic modulation by electric field. The mechanism described in Figure 5 with additional supporting information such as XPS to confirm the change in the percentage of oxidation state, and MFM to confirm the expanding FM domain expansion/shrinkage under electron doping/depletion would be interesting. In the view of following comments, and revision the current work can be considered

Title: The title is more general, and has to be specific as the "hybrid capacitance" approach is applied for conductive LSMO under investigation.

Abstract: development of new-generation remove "comma" in Abstract

Article

- A common and direct approach to realize the magnetoelectric (ME) effect is via manipulation of the electronic structure of a magnetic material upon voltage-induced charge doping. (No reference)
 - To a first approximation, the sheer strength of the ME coupling is directly related to the concentration of the accumulated charge carriers that can switch or modify the interactions between magnetic moments. Therefore, a magnetic body with a high capacitance, i.e., the ability to store electrical charge, should lead to an enhanced ME effect. (No reference, and too much descriptive. It is well established that capacitance plays a important role as described in reference 2)
- Does this "hybrid supercapacitor" approach to control magnetism feasible for insulating magnetic phases?
- What is the effect of larger thickness of LSMO for modulating its magnetism by "hybrid supercapacitive" approach?
- Figure 1 : writings in Figure 1 not clear
- Figure 2(a) mentions potential window ($0:2 \text{ V} < \Delta V < 3:7 \text{ V}$) but x-axis shows only up to 2 V
- the biggest applied potential window, the surface charge modulation reached a value of 270 (what was the biggest applied potential?)
- Thanks to surface charge densities as high as $\approx 270 \mu\text{C}/\text{cm}^2$ 95, we comprehensively explore the surface magnetoelectronic phase diagram of LSMO. thanks to the relatively small external voltage chosen (Why so much thanks for scientific experiments ?)

Reviewer #2 (Remarks to the Author):

I have read the manuscript "Hybrid supercapacitors for reversible control of magnetism" by Alan Molinari and collaborators submitted to Nature Communications for consideration. The paper reports on the control of the magnetism of an LSMO layer by means of electrostatic and electrochemical doping with an ionic liquid. The LSMO is the electrode of a supercapacitor with a DEME- TSFI ionic liquid electrolyte, magnetic moment is measured with a SQUID magnetometer, while the supercapacitor is polarized. The manipulation of surface magnetism with electric gating has been reported before in several publications. See Phys. Rev. Lett. 113, 267202 with a solid electrolyte, and Adv. Mater. Interfaces 2015, 2, 1500407 also in manganites and with an ionic liquid electrolyte. In my opinion the results lack of the novelty and significance to be published in Nat Comm. Also there are technical issues (see below) which per se disqualify publication in this or in any other journal. My recommendation is not publishing this paper.

The first problem is the magnitude of the effect. It is very small. The largest magnetization change detected is 2% according to the scale bar of figure 4 b. Small changes of magnetization are to be expected since as discussed by the authors the doping process occurs only at a small distance from the surface (Thomas Fermi screening). The thickness of the sample used is too large (13 nm) compared with the screening length. As a result the relative changes caused by doping on extensive quantities, such as it is the magnetic moment, are small. The absolute values of the moment change (Δm) are not quoted. The detection of signals close to the detection limit of a SQUID magnetometer in which electrical leads are feed down to the sample has to be demonstrated, as they may be an electric artifact.

The nature of the doping process in the electrochemical reaction regime is not discussed. What is the result of the electrochemical reactions detected by voltametry/capacitance measurements? Can it be probed with spectroscopic tools? Also, the film has a depressed magnetization as shown in the supplementary materials section, but usually depressed magnetization is due to the dead layer at the substrate. How do authors know that the surface of the LSMO film is phase separated? Unless these questions are addressed, the model proposed to discuss the non monotonic effect of the electron doping is pure speculation.

Reviewers' Comments:

Reviewer #1 (Remarks to the Author):

In the current work, involving hybrid supercapacitance approach to control magnetism in LSMO is interesting, and the results presented are efficient compared to traditional way of using only EDL capacitance. The experimental results confirm the significant role of pseudo capacitance by creating large surface charge density in comparison with EDL capacitance as described in Figure 1 and 2. Further in Figure 3 and 4, surface charge modulation mechanism in LSMO is elaborated by studying $M(T)$ under constant charge modulation, and varying constant charge modulation under isothermal condition to insight on mechanism of magnetic modulation by electric field. The mechanism described in Figure 5 with additional supporting information such as XPS to confirm the change in the percentage of oxidation state, and MFM to confirm the expanding FM domain expansion/shrinkage under electron doping/depletion would be interesting. In the view of following comments, and revision the current work can be considered.

We are pleased about the positive feedback and highly appreciate the detailed comments and suggestions for the improvement of the manuscript. We agree with the reviewer that additional information regarding the change in manganese oxidation state and the expansion/shrinkage of the FM domains would be very interesting. Nonetheless, for this purpose, it would be necessary to perform very challenging *in-situ* spectroscopic measurements capable of penetrating the ionic liquid and delivering information about the processes occurring at very surface of LSMO while applying an external voltage. We hope to be able to do these beautiful and non-trivial experiments in a future study (or maybe other groups will be willing to tackle this challenge). Here, in our manuscript, we rather aim to report about the presence of surface pseudocapacitance and its remarkable effect on the magnetic properties.

- Title: The title is more general, and has to be specific as the “hybrid capacitance” approach is applied for conductive LSMO under investigation.

We have to admit that the choice of a rather general title was intentional (we also felt that a more general aspect and not specialized is what Nature is looking for. Indeed, we received rejections in the past on other manuscripts because the editors felt that the subject was not general enough). We think that the effect of surface pseudocapacitance to control magnetism is a general concept and will work not only for LSMO, but also for many other classes of systems where magnetic states and properties are determined by the charge carrier concentration. For this reason we think that it is more appropriate to formulate the title in wide-ranging terms. For example, it would be interesting to take a closer look at the cobaltites (e.g. LaSrCoO₃) and ferrites (e.g. LaSrFeO₃) and diluted magnetic semiconductors. In this respect we have modified a sentence in the conclusions [lines 425-429].

- Abstract: development of new-generation remove “comma” in Abstract

We removed the comma [line 11]

Article

- A common and direct approach to realize the magnetoelectric (ME) effect is via manipulation of the electronic structure of a magnetic material upon voltage-induced charge doping. (No reference)

We think that Ref. 2 (Matsukara *et al.*, Nat. Nano. 10 (2015)) may be an appropriate reference for this statement, and it is now referenced at the end of the sentence [line 29]. Their work is a review about the possible ways to control magnetism in semiconductors, metals and multiferroics, and may be useful for the reader who wants to know more about the topic.

- To a first approximation, the sheer strength of the ME coupling is directly related to the concentration of the accumulated charge carriers that can switch or modify the interactions between magnetic moments. Therefore, a magnetic body with a high capacitance, i.e., the ability to store electrical charge, should lead to an enhanced ME effect. (No reference, and too much descriptive. It is well established that capacitance plays an important role as described in reference 2)

We suggest to explain the concept in the following more compact way: "It has been experimentally shown that the strength of the ME effect at the interface is directly related to the electrical capacitance (Matsukara *et al.*, Nat. Nano. 10 (2015); Leufke *et al.*, Phys. Rev. B. 87 (2013)),, [lines 43-45]

- Does this "hybrid supercapacitor" approach to control magnetism feasible for insulating magnetic phases?

In order to bring the charge at the interface between an ionic liquid and a magnetic material, the latter must be able, at least partially, to conduct charge carriers. In this sense, even if a material is a poor conductor, it should be possible to charge it under certain conditions. For example, if we consider a capacitor as made of an ideal capacitor (with capacitance C) in series with a high resistor (with resistance R), it is still possible to charge the capacitor if sufficient time is given for the charging process. This specific charging time is defined by the time constant $\tau = RC$ of the circuit.

Giving an example from another research area, Ueno *et al.*, Nat. Mat. 7 (2008) were capable of inducing superconductivity in an insulating single-crystal of SrTiO₃ by charging it with an ionic liquid. In that case the authors attributed the effect to pure electric-double-layer capacitance.

Moreover, we can provide also evidence from our own experiments. Recently, we performed cyclic voltammetry measurements on our LSMO films above the Curie temperature (where it is in a paramagnetic and nominally insulating state). Although LSMO behaves as a very poor conductor (or almost insulator) it is still possible to observe the typical response of a capacitor in the current-voltage curves.

These examples strongly suggest that "hybrid capacitance" may also be used to control magnetism of insulating or poorly conducting magnetic phases. Actually, our work is supposed to stimulate further studies in this direction.

- What is the effect of larger thickness of LSMO for modulating its magnetism by “hybrid supercapacitive” approach?

Since our tuning approach exploits surface effects, a decrease in the relative magnetic modulation is expected upon increasing the LSMO thickness.

In the future it would be interesting to use the effect of “hybrid capacitance” to study thinner samples. By increasing the surface-to-volume ratio we expect an increase in the magnitude of the magnetic modulation. However, it is also known (as reported for example in Huijben *et al.*, Phys. Rev. B 78 (2008)) that by decreasing the LSMO thickness, the para-ferromagnetic transition temperature decreases as well. At the moment we are satisfied with the choice of a thickness of 13 nm, because the magnetic transition is around 323 K. In this way we demonstrated a magnetic tuning effect of up to 33% around room temperature, which is always desirable in the perspective of possible future applications.

- Figure 1 : writings in Figure 1 not clear

We have modified the caption of Fig. 1 in a clearer manner.

- Figure 2(a) mentions potential window ($0.2 \text{ V} < \Delta V < 3.7 \text{ V}$) but x-axis shows only up to 2 V

We think that at this point there is some technical misunderstanding. There are negative and positive values of voltage in the x-axis of Fig.2a and the potential window is given by the difference between maximum and minimum voltage $|\Delta V| = V_{\max} - V_{\min}$, which is varied in the range of $0.2 < |\Delta V| < 3.7 \text{ V}$.

- the biggest applied potential window, the surface charge modulation reached a value of 270 (what was the biggest applied potential?)

We have obtained a surface charge modulation of $270 \mu\text{C}/\text{cm}^2$ using a potential window of 3.7 V. The specific value is now expressly stated in the manuscript [line 188].

- Thanks to surface charge densities as high as $\approx 270 \mu\text{C}/\text{cm}^2$, we comprehensively explore the surface magnetoelectronic phase diagram of LSMO. thanks to the relatively small external voltage chosen (Why so much thanks for scientific experiments ?)

We have adjusted the two sentences as following:

“We exploit surface charge densities as high as $\approx 270 \mu\text{C}/\text{cm}^2$ to comprehensively explore the magnetoelectronic phase diagram of LSMO.” [lines 90-92]

“These findings give a strong indication that under application of small external voltages, the CV measurements were performed within a regime dominated by electrostatic doping.” [lines 161 – 165]

Reviewer #2 (Remarks to the Author):

I have read the manuscript “Hybrid supercapacitors for reversible control of magnetism” by Alan Molinari and collaborators submitted to Nature Communications for consideration. The paper reports on the control of the magnetism of an LSMO layer by means of electrostatic and electrochemical doping with an ionic liquid. The LSMO is the electrode of a supercapacitor with a DEME- TSFI ionic liquid electrolyte, magnetic moment is measured with a SQUID magnetometer, while the supercapacitor is polarized. The manipulation of surface magnetism with electric gating has been reported before in several publications. See Phys. Rev. Lett. 113, 267202 with a solid electrolyte, and Adv. Mater. Interfaces 2015, 2, 1500407 also in manganites and with an ionic liquid electrolyte. In my opinion the results lack of the novelty and significance to be published in Nat Comm. Also there are technical issues (see below) which per se disqualify publication in this or in any other journal. My recommendation is not publishing this paper.

We would like to thank the reviewer for the time devoted to reading our manuscript. However, we feel that we have to disagree with him on the major points of his criticism.

In our manuscript we do not claim to be the ones to introduce the general concept of manipulating surface magnetism with ionic liquid (or electrolyte) gating. The gist of our manuscript is to show that:

1. Surface pseudocapacitance is a new tool to reversibly and robustly control magnetism. The effect is not related to the well-known electric-double-layer or ionic intercalation mechanisms to tune magnetism.
2. At a fixed temperature we prove to control the magnetic response in-phase and/or anti-phase (see Fig. 4) with respect to the surface charge modulation by simply adjusting the external voltage. This level of flexibility in controlling magnetism has not been achieved so far.
3. We modulate the magnetization of LSMO up to a maximum of 33 % (Fig.6) around room temperature. To date, this is the highest value reported around room temperature for electrolyte-gated manganites. Furthermore we demonstrate that the effect is fully reversible.

We have read the work of Bi *et al.*, Phys. Rev. Lett., 113 (2014) and Ge *et al.*, Adv. Mat. Interf., 2, (2015), mentioned by the reviewer. In this case rather than “surface” magnetism we would use the term “bulk” magnetism, since both works deal with the intercalation of oxygen ions through interfaces and diffusion into the bulk of a film to control the transport and magnetic properties.

Let us briefly comment on the work of Ge *et al.*, who studied the transport (and not magnetic) properties of a LSMO film gated with an ionic liquid under application of an external voltage. Looking at their TEM images after application of 2.5 V for 90 min (Fig 2a,b in Ge *et al.*), the authors state that “*The depth of oxygen vacancies induced by IL gating reached a dozen of nanometers from the surface of the LSMO film under such conditions*”. This means that not only the surface, but also the bulk of the film participate in the oxygen diffusion. Also, we are in full agreement with the following conclusions of the authors:” *The present work combined with the recent studies indicates that there may be several mechanisms behind the electronic phase modulation related to electrolyte gating in oxides. Which one dominates in these phenomena possibly depends on both the treatment of electrolyte and the choice of gated materials. There are actually electrochemical reactions using water contaminated IL as gating electrolyte, and more oxygen vacancies can be produced with more water in IL. This electrochemical mechanism can be greatly suppressed by exploiting*

dry IL. Nevertheless, the strong electric field at the interface between IL and oxides almost cannot induce oxygen vacancies in LSMO, different to the case in VO₂".

Here, we would like to state the reasons why, differently from Bi et al. and Ge et al., we have dismissed oxygen intercalation as a major contributor to the ME coupling in our LSMO/ionic liquid system:

- We use a highly purified ionic liquid, with a water content < 50 ppm, as measured via Karl-Fisher titration (as stated in Methods). Thus, the low amount of oxygen or OH groups produced by decomposition of contaminating water cannot be a major contributor in controlling magnetism.
- It is known that water decomposition starts to occur for applied voltages > 1.2 V. Most of our data (see Supplementary Tables 1, 2, 3) are well below this threshold voltage.
- Let us assume that for some unknown reasons our highly-purified ionic liquid contained a considerable amount of oxygen ions. For thermodynamic and kinetic reasons (consider for example the Nernst-Einstein relation in presence of electric fields) in order to allow oxygen ions to intercalate and diffuse through an interface, it is necessary to overcome a certain energy barrier, which depends on applied voltage, temperature, time, kind of involved species,... (e.g. Subramanian et al., *Ener. Envir. Sci.* 2 (2009)). Considering the papers mentioned by the reviewer, Bi et al., applied a constant voltage of ± 5 V for 13 min at 200 °C or ± 5 V for about 30-150 sec at 260 °C, while, Ge et al., applied a constant voltage of 2.5 V for 15 - 265 min. These high values of voltage, temperature and voltage-holding time are not compatible with our cyclic voltammetry measurements. The maximum voltage that we applied is about ± 1.8 V at the low temperature of -53 °C (220 K) and, in general, all our measurements are performed by continuously sweeping the voltage (typically with a scan rate of 3 mV/s). The highest voltage that we applied at room temperature was ± 0.9 V, which is anyway below the water decomposition voltage of 1.2 V.
- We have carefully analyzed the behavior of the charging current as a function of the voltage ramp rate (Fig. 2c) for fixed potential windows. It is known (see for example Augustyn *et al.*, *Energ. Environ. Sci.* 7 (2014) or Brezesinski *et al.*, *Nat. Mat.* 9 (2010)) that a linear behavior of the current indicates the presence of fast surface charging/discharging processes (as expected for the case of electric-double layer and pseudocapacitive charging). This is exactly the behavior shown in Fig. 2c. A square root behavior instead would mean the presence of diffusion/intercalation into the bulk of LSMO, but this is clearly not observed in the case of our LSMO.
- We are aware of the fact that minor water contamination in ionic liquids can be reduced but cannot be completely avoided. We have indication of some minor oxygen-diffusion effects also in our experimental results: this can be observed in Fig. 2a, which displays the presence of some small irreversible peaks close to -1 V in both positive and negative charging current directions. As this is a minor effect, we did not deeply discuss it in the manuscript, but we only mentioned that "*At a certain limit, when $|\Delta V|$ was considerably expanded up to 3.7 V, small irreversible redox peaks started to appear as well (see arrows in Fig. 2a)*" [lines180-183].

Summarizing: In contrast to the works referred to by the reviewer in our case oxygen diffusion cannot be a major factor in controlling magnetism, because we use low voltages, low temperatures, we do not apply a constant voltage and, in general, we use a highly purified ionic liquid.

We argue that the main phenomenon driving the control of magnetism is surface pseudocapacitance and not oxygen diffusion into LSMO. Our experimental results indicate that, upon application of small external

voltages, the DEME/TFSI ions are physisorbed onto the LSMO surface, forming the well-known Helmholtz double layer. In this regime, electrostatic doping is the main mechanism inducing the accumulation of charge carriers at the LSMO surface. By applying higher and higher voltages, the ions experience stronger and stronger adhesive forces, which gradually bring to ionic chemisorption onto the LSMO surface. This new regime is dominated by surface redox reactions where charge carriers (faradaic current) are exchanged between the ionic liquid and LSMO. The effect on magnetism of LSMO is due to charge carrier doping and not to the removal/insertion of oxygen ions. The phenomenon that we observe should not be confused with the well-known electrostatic doping effect occurring in electric-double layer capacitors. An in-depth discussion of the pseudocapacitive phenomena, their physic-chemical classification and specific experimental examples are given, for example, in Conway *et al.*, J. Power Sources (1997) and Conway *et al.*, J. Electrochem. Soc. 138 (1991).

According to the established procedures, our cyclic voltammetry data give unequivocal proof for the presence of surface pseudocapacitance. To our knowledge, surface pseudocapacitance (differently from oxygen intercalation or electrostatic doping) has never been used to control magnetism. This is one of the novelties of this work, which aims at merging the broad research fields of magnetoelectrics and supercapacitors.

Coming back to the manuscript, we have clarified some of the above discussed points in the section “Capacitive and Pseudocapacitive charging regimes” [see lines 195-200]. We have considered the possible influence of oxygen diffusion into LSMO and the reasons for its negligible role in our case. We have cited the work of Ge *et al.*, because it is absolutely pertinent to the discussion. If the reviewer prefers, we can also cite the paper of Bi *et al.*, but probably it should be rather inserted in the introduction of the manuscript.

- The first problem is the magnitude of the effect. It is very small. The largest magnetization change detected is 2% according to the scale bar of figure 4 b. Small changes of magnetization are to be expected since as discussed by the authors the doping process occurs only at a small distance from the surface (Thomas Fermi screening). The thickness of the sample used is too large (13 nm) compared with the screening length. As a result the relative changes caused by doping on extensive quantities, such as it is the magnetic moment, are small.

We are a bit baffled by some of the comments and do not agree with the suppositions of the reviewer. It is stated, for instance already in the abstract (but please, check also Fig. 6a and the discussion), that we have achieved a relative modulation in magnetization of up to 33 %, which is an order of magnitude higher than the value the reviewer refers to. When the reviewer refers to the 2 % effect in Fig. 4b, it should be underlined that this value was obtained at the low temperature of 220 K. The reason for the small effect at low temperature is that LSMO is far from the para-ferromagnetic phase transition: LSMO is predominantly metallic (see also Ge *et al.*,) and so, indeed, the low penetration depth of the electric field allows for probing only about one LSMO unit cell (e.g. Molegraff *et al.*, Adv. Mater. 21 (2009)). As the reviewer pointed out, the value of 2 % obtained at 220 K is present in the scale bar of Fig. 4b and it is not explicitly mentioned in the manuscript, but, nevertheless, we clearly state: “Furthermore, it is worth underlying that the small $\Delta M/M$ ratio encountered in all the measurements corroborates the idea that tuning is a surface effect, as already concluded.” [lines 309-312]. However, by increasing the temperature up to 323 K, which is close to the para-ferromagnetic phase transition, we are able to reach a $\Delta M/M$ of about 33 %. To our knowledge, this is also the highest value so far reported around room temperature for electrolyte-gated manganites films; further the effect is reversible upon cycling the voltage (see Fig. 6a).

It is worth stressing that the purpose of Fig. 4. is not to show the magnitude of the effect, but to demonstrate that we can modulate the magnetization of LSMO *in-phase and or anti-phase* with respect to the surface charge (induced by the external voltage). This is another focal point of this paper: to our knowledge it has never been achieved this level of freedom in controlling the magnetic response at a fixed temperature by simply adjusting the external voltage. Further, by the slopes of Fig. 4b, it is possible to quantitatively calculate the values of magnetoelectric coupling coefficient $|\alpha| = |\Delta M/\Delta Q|$.

In general, we think that not only the maximum magnitude of the effect, despite of its remarkable value, should be taken into account, but also the novelty of the basic physical principles, which, once understood, will allow for increasing the effect magnitude for example by investigating other materials.

The use of a LSMO film with a thickness of 13 nm and not an ultrathin film is positively intentional. We wanted to have the para-ferromagnetic phase transition around room temperature (which is always desirable in the perspective of possible future applications). It is known that upon decreasing the thickness of LSMO films, the Curie temperature decreases well below room temperature (see for example Huijben *et al.*, Phys. Rev. B 78 (2008)).

- The absolute values of the moment change (Δm) are not quoted. The detection of signals close to the detection limit of a SQUID magnetometer in which electrical leads are feed down to the sample has to be demonstrated, as they may be an electric artifact.

Here the reviewer, for all the practical purposes, questions all our experimental work on the, rather vague grounds that "*they may be an electric artifact*". The reviewer argues that the measurements were done at the detection limit of the SQUID.

Here we take a liberty of strongly disagreeing with the reviewer's supposition. Of course, over the years we have developed a rigorous protocol of *in situ* SQUID experiments to exclude any artificial contributions. Actually the moment change (ΔM) is quoted in the manuscript (see Fig. 3 and Fig.4), but it is normalized to the surface unit cell to account for the fact that charge accumulation occurs at the LSMO surface. When converted to the actually measured values, we obtain a magnetization change in the range of 10^{-7} - 10^{-6} emu; the sensitivity of our SQUID magnetometer is $< 10^{-8}$ emu. The decisive factor in our measurement procedure is that we record the change of the magnetic moment during the charging; this means that any extra contributions to the total magnetization coming from the substrate, chemical cell, wiring etc. are automatically factorized out allowing for taking full advantage of the SQUID ($< 10^{-8}$) emu sensitivity. In this experimental approach we definitely do not operate at the SQUID detection limit.

As to the possible contribution of the charging current to the measured ΔM , there is a very fundamental test to exclude the artificial effects. One can do charging experiments for two orientations of the magnetization by simply changing the direction of the magnetic field. In this case the genuine magnetoelectric effect will change its sign together with the changed magnetization (following external field). But any artificial contribution to magnetization coming from the charging current will not change its direction at all. We have run this test on our samples and we have never seen any trace of the charging current-induced artificial magnetization.

Of course, if necessary, we would be more than happy to share more details of these tests with the reviewer. More importantly, however, we have already shown in our previous works (Leufke *et al.*, Phys. Rev. B., 87 (2013); Mishra *et al.*, J. Appl. Phys. 113 (2013); Reitz *et al.*, Chem. Mater. 26 (2014)) that even more subtle and demanding magnetoelectric effects in manganites can be measured unambiguously using our *in-situ* setup.

- The nature of the doping process in the electrochemical reaction regime is not discussed. What is the result of the electrochemical reactions detected by voltametry/capacitance measurements? Can it be probed with spectroscopic tools?

As mentioned above, several works dealing with in-depth discussions of the mechanisms behind pseudocapacitive phenomena, their physic-chemical classification and specific experimental examples can be found in the literature. In our manuscript we have referenced [see Ref 19-22] some of the most important and cited works in the field of pseudocapacitors. Our aim in the manuscript is to discuss about the influence of the huge accumulated surface charge by means of surface pseudocapacitance on the magnetic properties, rather than focusing on a deeper understanding of the underlying interfacial charging mechanisms (but of course we hope that our work will trigger further studies in this direction).

Furthermore, frankly speaking, we have trouble to envision an experimental setup providing the appropriate spectroscopic tools. Indeed it would be necessary to perform *in situ* spectroscopic measurements capable of penetrating the ionic liquid and revealing changes in the oxidation state at the very surface of LSMO while applying an external voltage. This displays a very interesting as well as non-trivial experiment, which goes far beyond the purposes of the present manuscript, but we hope to be able to tackle this challenge in the future. As a comparison to the work mentioned by the reviewer, Ge *et al.*, extracted/inserted oxygen ions into and out from LSMO using a voltage and, afterwards, they performed *ex situ* TEM measurements in order to analyze the profile of oxygen vacancies throughout the LSMO film.

- Also, the film has a depressed magnetization as shown in the supplementary materials section, but usually depressed magnetization is due to the dead layer at the substrate.

We do not fully understand the purpose of this comment; we are aware of the existence of the magnetic dead layer as clearly stated in the Supplementary Information: "*The theoretical bulk value⁵ of $M_{sat} = 3.74 \mu_B/u.c.$, usually not reachable in thin films, can be obtained under the assumption of a magnetically dead layer of about 3.5 nm at the LSMO/STO interface, a rather reasonable value comparing to other literature reports⁶*".

- How do authors know that the surface of the LSMO film is phase separated? Unless these questions are addressed, the model proposed to discuss the non monotonic effect of the electron doping is pure speculation.

The phenomenon of magnetic (and also electronic) phase separation in manganites has been known for more than 15 years. In the literature a huge number of reports can be found about phase separation in manganites and more specifically in LSMO; for example we kindly invite to check Moreo *et al.*, Science 283 (1999); Becker *et al.*, Phys. Rev. Lett. 89 (2002); Fäth *et al.*, Science 285 (1999). Actually, phase separation in LSMO is also mentioned in Ge *et al.* (Adv. Mater. Interfaces 2015, 2, 1500407), which is the work remarked by

the reviewer in his comments above. Further, we point out that the control of phase separation in manganites by means of an electric field has already been used by Lourembam *et al.*, (Phys. Rev. B, 89 (2014)) in Pr(CaSr)MnO₃ films gated with an ionic liquid.

Without making use of the concept of phase separation, there is no other feasible way to explain properly our quantitative results which give a magnetoelectric coupling coefficient of $|\alpha| = |\Delta M/\Delta Q| \sim 3 \mu_B/h^+$ at 220 K. The classical interpretation is that at low temperature LSMO is in the metallic phase: thus each charge carrier that is inserted at the LSMO surface under application of the external voltage should contribute to the magnetization with $\alpha = -1 \mu_B/h^+$ (or $+1 \mu_B/e^-$), as in accordance with a “band filling” model. Our precisely measured value of $|\alpha| = |\Delta M/\Delta Q| \sim 3 \mu_B/h^+$ is way too high and gives strong indication that each introduced charge carrier rather switches on/off the entire magnetic moment carried by an entire LSMO unit cell, i.e. $M_{Mn}=(4-x) \sim 3.7 \mu_B/u.c.$ Please, we invite the reviewer to refer to Molegraft *et al.*, Adv. Mater. 21, (2009), who also discussed about the failure of the classical “band filling” model.

Regarding the manuscript, we hope to have provided enough clarifications about our *in situ* SQUID-cyclic voltammetry results. If there are still some open questions we will be happy to share additional details about the experimental setup and procedures employed during the experiments.

Reviewer #2 (Remarks to the Author):

I have read the revised manuscript "Hybrid supercapacitors for reversible control of magnetism" by Alan Molinari and collaborators, as well as authors response to my criticism to their first version. Authors have made an effort to address my criticism and made revisions to the manuscript. Although I consider the results of interest, I maintain my view that the paper does not represent an advance in the field of enough significance to be published in Nature Communications. The reasons, after reading the documents submitted by the authors, are basically similar to those contained in my previous report and can be summarized as follows:

1) The paper reports on the control of the magnetism of an LSMO layer which is the electrode of a supercapacitor with a DEME- TSFI ionic liquid electrolyte. Changes in magnetism are detected through measurements of the magnetic moment of a 13 nm thick layer which is being modified by electrochemical reactions only in its surface. As a result, the largest moment changes detected (close to the Curie temperature) are below 4%. These are then scaled by authors to 33 % of the surface moment assuming that only the surface unit cells are being modified. However this is arbitrary. The Thomas Fermi length in oxides is typically much closer to 1 nm in oxides. If authors want to make a claim that surface magnetism is modified, the experiment should be capable of measuring the thickness of such "surface" layer (possibly by reducing the thickness of the LSMO sample).

2) Authors put forward an explanation based on electrically controlled magnetoelectronic phase separation based on the modification of the magnetic interaction between Mn neighbors from ferromagnetic to antiferromagnetic. This is also purely speculative. There is not a single proof in the paper on the laterally inhomogeneous change of the oxidation states nor on the modified magnetic interactions between neighbors. As I said in my previous report there is no spectroscopic diagnostics of the effect of the electrochemical reactions responsible of the modified magnetism.

In summary, I do not consider that the claim that "surface pseudocapacitance is a new tool to reversibly and robustly control magnetism" made in this paper is actually supported by the experimental data and also not that it is different in nature to the electrostatic or electrochemical doping at other ionic liquid /oxide doping experiments.

Reviewer #3 (Remarks to the Author):

The manuscript by Hanh et al. describes an interesting study that uses electrochemical methods to modulate non-faradaic electrochemical double layer capacitance and faradaic pseudocapacitance in LSM/ionic liquid interfaces to control the balance between ferromagnetic and non-ferromagnetic phases.

This paper should be published as a whole.

I have only a few helpful considerations.

There are several ex situ and in situ xrd studies of phase transitions of perovskites and ruddlesden-popper phases that under go electrochemically induced intercalation, redox and phase change upon applied bias. It would be nice if such data was available for this work. As the authors cite recent studies on LMO demonstrate that nanoscale sized particles can support high electrochemical pseudocapacitance via oxygen intercalation. In this paper it is not clear as to the role of oxygen vacancies relative to magnetic ordering effects. As in the cited Mefford paper...the higher the oxygen vacancy concentration the higher the pseudocapacitance based on their proposed mechanism. As such how does this vacancy (substoichiometry) it should have a big influence their claims of magnetic ordering and switching? The authors also should note that there is recent theoretical work that aims to correlate oxygen evolution activity with magnetic properties. J. Garcia et al ChemCatChem 2016, 8, 2968 – 2974. While the authors are not experts in this field they should be aware that there should be a strong link between pseudocapacitance, oxygen intercalation, intervalance charge transfer rxns and other oxygen based reactions. There is also some other work in intercalation based superconductivity switching.

It is not clear why the authors chose such a specific ionic liquid vs other media. Their electrolyte is somewhat exotic and as perhaps others have commented hard to make water and oxygen free which could bias some of their measurements. This reviewer is curious as to if such magnetic effects can be evidenced in strong basic electrolytes relevant to water oxidation and anion based pseudocapacitance. Perhaps theory could assist in that this choice of ionic liquid creates a very special field assisted ordering of the double layer of ions which also influences magnetic ordering. To my knowledge there are no examples of such experiments or calculations to support this concept. However, perhaps these results provide impetus for new paradigms for understanding such complex behaviors in more complex interfacial processes. Most experiment and theory is stuck on single metal solid/liquid interfaces and have yet to evolve to well controlled metal oxide electrochemical interfaces.

As such this paper stands on its own merit and should be published. yet much work is ahead to understand complicated chemical and physical mechanisms of complex metal oxides.

Reviewers' Comments:

Reviewer #2 (Remarks to the Author):

I have read the revised manuscript “Hybrid supercapacitors for reversible control of magnetism” by Alan Molinari and collaborators, as well as authors response to my criticism to their first version. Authors have made an effort to address my criticism and made revisions to the manuscript. Although I consider the results of interest, I maintain my view that the paper does not represent an advance in the field of enough significance to be published in Nature Communications. The reasons, after reading the documents submitted by the authors, are basically similar to those contained in my previous report and can be summarized as follows:

1) The paper reports on the control of the magnetism of an LSMO layer which is the electrode of a supercapacitor with a DEME- TSFI ionic liquid electrolyte. Changes in magnetism are detected through measurements of the magnetic moment of a 13 nm thick layer which is being modified by electrochemical reactions only in its surface. As a result, the largest moment changes detected (close to the Curie temperature) are below 4%. These are then scaled by authors to 33 % of the surface moment assuming that only the surface unit cells are being modified. However this is arbitrary. The Thomas Fermi length in oxides is typically much closer to 1 nm in oxides. If authors want to make a claim that surface magnetism is modified, the experiment should be capable of measuring the thickness of such “surface” layer (possibly by reducing the thickness of the LSMO sample).

We would like to thank the reviewer for taking the time to read our response to his criticisms. However, much to our regret, we must disagree once again with the reviewer’s comments. The largest change of the magnetization, $\Delta M / M$, detected upon electric charging (close to the Curie temperature) is 33%. This is an experimentally measured value without any sort of rescaling; ΔM is simply the total magnetization change and M is the total magnetization measured at the temperature of interest. It is a natural and very straightforward measure of the effect and, in our opinion, there is nothing arbitrary about it.

Perhaps the reviewer’s misunderstanding might arise from the two different representations of ΔM used in the manuscript; in one case we normalize the magnetization change to the surface unit cell (ΔM_o ($\mu_B/u.c.^2$)) in the other to the total bulk, untuned LSMO magnetization ($\Delta M/M$ (%)) as specified in the lines 242-247. To avoid any possible unambiguity, the authors have rewritten the definitions of ΔM_o and $\Delta M/M$ in a clearer way (lines 242-247). Furthermore, we have clarified that the small $\Delta M/M$ values were found for the low-temperature measurements, while a bigger effect was encountered in proximity of T_c (lines 309 -315).

We agree that often in the literature the Thomas-Fermi screening length for conductive oxides is assumed to be around 1 nm (or even less). However, this pertains to the one-phase, homogeneous metallic systems. This simple, purely metallic, picture breaks down when the magneto-electronic phase separation into metallic and insulating is considered. There are many experimental and theoretical works indicating that the electric field can effectively penetrate much deeper into phase-separated manganite films. For example Dhott et al., Phys. Rev. Lett. 102 (2009) demonstrated that in electrolyte-

gated LaBaMnO₃ films the screening length can extend up to 5 nm when the system is close to the critical temperature. Interestingly, in proximity of T_c they obtain a change in resistivity of $\Delta R/R = 29\%$ for a 11 nm LaBaMnO₃ film, while we obtain a change in magnetization of $\Delta M/M = 33\%$ for a 13 nm LaSrMnO₃ film. The similarity in the size of the two effects is quite astonishing, but reasonable, if considering that magnetic and transport properties in manganites are strongly correlated. Furthermore, the phenomenon of phase separation provides another indication that the screening length is not constant but changes drastically as a function of temperature. For example, Becker et al., Phys. Rev. Lett. 89 (2002), revealed via scanning tunneling microscopy that below T_c LaSrMnO₃ and LaBaMnO₃ films present insulating and metallic domains, whose size shrinks or expands as a function of temperature (with more insulating areas at higher temperatures and viceversa).

We decided to strengthen this point by adding a few sentences at the end of the manuscript where we report of $\Delta M/M = 33\%$ in proximity of T_c (lines 401-412).

2) Authors put forward an explanation based on electrically controlled magnetoelectronic phase separation based on the modification of the magnetic interaction between Mn neighbors from ferromagnetic to antiferromagnetic. This is also purely speculative. There is not a single proof in the paper on the laterally inhomogeneous change of the oxidation states nor on the modified magnetic interactions between neighbors. As I said in my previous report there is no spectroscopic diagnostics of the effect of the electrochemical reactions responsible of the modified magnetism.

The interpretation of electrically controlled magnetoelectronic phase separation is based on the following facts: First, our experimental results at 220 K for various charge doping concentrations unequivocally provide a magnetoelectric coupling coefficient $|\alpha| \sim 3 \mu_B/h^+$. If a classical band-filling model were correct, one should expect that each introduced carrier contributes to the magnetization with $1 \mu_B$ (one more electron = one added spin, one more hole = one removed spin). Our value is about 3 times larger and rather close to the magnetic moment carried by an entire LSMO unit cell ($M_{Mn} \sim 3.7 \mu_B/u.c.$). If we assume that the LSMO surface is phase separated (as already shown by other groups), it becomes rather reasonable envisioning a scenario where some magnetic domains grow at the expense of others by incorporating one entire LSMO unit cell for each charge carrier introduced by doping. Second, we are not the first ones who claim the usage of an electric field to control phase separation in manganites, as for example in the paper of Lourembam et al., Phys. Rev. B. 89 (2014), entitled "Electric field tuning of phase separation in manganite thin films". Third, phase separation in LSMO has already been shown in both theoretical (for ex. Moreo et al., Science 283 (1999)) and experimental (Becker et al., Phys. Rev. Lett. 89 (2002) works.

Summarizing, the results reported in literature for phase-separated manganites and our values of magnetoelectric coupling coefficient provide strong indication of the tuning of magnetoelectronic phase separation using an external electric field. We have clarified the discussion about magnetoelectronic phase separation in a few points (lines 335-342, 359-361 and 385).

We do not agree with the reviewer when he states that there is not a single proof on the modified magnetic interactions between neighbors. It is well established (e.g. Coey et al., Adv. Phys. 58 (2009)) that magnetism in LSMO is driven by the competition between double and superexchange interactions, promoting ferromagnetic and antiferromagnetic coupling between adjacent Mn ions, respectively. Our results clearly show that magnetism “follows” the charge added/removed at the LSMO/ionic liquid interface, thus, as a consequence, the double and superexchange interactions between neighbors Mn ions must be affected.

As we already wrote in our previous response, it is definitely a formidable task to envision an experimental setup to perform *in situ* spectroscopic measurements capable of penetrating the ionic liquid and revealing changes in the oxidation state at the very surface of LSMO while applying an external voltage. This goes far beyond the purposes of the present manuscript, nonetheless we hope to be able to tackle this challenge in the future.

In summary, I do not consider that the claim that “surface pseudocapacitance is a new tool to reversibly and robustly control magnetism” made in this paper is actually supported by the experimental data and also not that it is different in nature to the electrostatic or electrochemical doping at other ionic liquid /oxide doping experiments.

To our knowledge, surface pseudocapacitance is a new tool to control magnetism because, so far only electric double layer charging and bulk ionic intercalation have been used to explain the control of magnetic and transport properties in manganites via external electric fields. Our cyclic voltammetry measurements unequivocally show that surface chemistry participate in the charging/discharging processes. Our SQUID measurements clearly show an outstanding control over the LSMO magnetization by exploiting both electrostatic and electrochemical doping: M can be modulated in-phase and/or antiphase with respect to Q , the magnetic tuning effect is up to $\Delta M/M = 33\%$ above room temperature (without any rescaling to the surface unit cell) and the tuning effect is highly reversible in a wide temperature region (220 – 330 K) and for different applied voltages (ΔV up to 3.7 V).

Reviewer #3 (Remarks to the Author):

The manuscript by Hanh et al. describes an interesting study that uses electrochemical methods to modulate non-faradaic electrochemical double layer capacitance and faradaic pseudocapacitance in LSM/ionic liquid interfaces to control the balance between ferromagnetic and non-ferromagnetic phases.

This paper should be published as a whole.

I have only a few helpful considerations.

There are several *ex situ* and *in situ* xrd studies of phase transitions of perovskites and ruddlesden-popper phases that under go electrochemically induced intercalation, redox and phase change upon applied bias. It would be nice if such data was available for this work. As the authors cite recent studies on LMO demonstrate that nanoscale sized particles can support high electrochemical pseudocapacitance via oxygen intercalation. In this paper it is not clear as to the role of oxygen vacancies relative to magnetic ordering effects. As in the cited Mefford paper...the higher the oxygen vacancy concentration the higher the pseudocapacitance based on their proposed mechanism. As such how does this vacancy (substoichiometry) it should have a big influence their claims of magnetic ordering and switching? The authors also should note that there is recent theoretical work that aims to correlate oxygen evolution activity with magnetic properties. J. Garcia et al ChemCatChem 2016, 8, 2968 – 2974. While the authors are not experts in this field they should be aware that there should be a strong link between pseudocapacitance, oxygen intercalation, intervalance charge transfer rxns and other oxygen based reactions. There is also some other work in intercalation based superconductivity switching.

We thank the reviewer for the time dedicated to reading our work and for the overall positive response.

In our understanding, *in situ* or *ex situ* x-ray diffraction (XRD) are suitable techniques to investigate the effects of bulk intercalation chemistry, where most of the material undergoes a phase change. Our approach relies on the control of magnetism via electrostatic and electrochemical doping primarily occurring at the very surface of the LSMO films. Since XRD is chiefly a bulk technique, we can hardly use it to analyze the surface tuning effect of our thin films. As to suitable spectroscopic techniques for *in situ* charging experiments, we face a general dilemma: on the one hand when the method has a right depth resolution to look at the surface effects (like XPS) then the presence of the ionic liquid on the top of the sample precludes its application; on the other hand when one can get through ionic liquid (like EXAFS, Neutron Scattering, XMCD) then the method of choice usually lacks the necessary depth resolution. Perhaps, some more sophisticated surface synchrotron spectroscopy techniques may provide the required sensitivity. Of course ways to get experimental insights into the local changes of oxidation state occurring at the LSMO/ionic liquid interface must be diligently pursued; however due to its inherent complexity this goes beyond the purposes of the present manuscript and will be definitely tackled in the future.

The approach of Mefford et al., is based on the intercalation of oxygen ions into the bulk of their $\text{LaMnO}_{3-\delta}$ powders, process that is facilitated by the presence of high oxygen vacancies concentration (intercalation pseudocapacitance). In our case, there is no intercalation but the ionic liquid gradually shifts from physisorption to chemisorption onto the LSMO surface upon increasing the external voltage (from electric-double-layer to surface pseudocapacitance). Thus, the control of the LSMO magnetic properties is reached by charge carrier doping, which modifies the oxidation state of adjacent Mn ions and, thus, also the magnetic exchange interactions ($\text{Mn}^{3+/4+}$ promotes ferromagnetic coupling, while $\text{Mn}^{3+/3+}$ or $\text{Mn}^{4+/4+}$ promote antiferromagnetic coupling). Hence, in our case it is not necessary to have oxygen deficiencies; actually we tried to reduce them via postannealing of our films for 1 h at 900 °C in air in order to obtain good magnetic properties of LSMO (Curie point and saturation magnetization close to bulk LSMO).

Nonetheless the remark of the reviewer is very appropriate. We strongly believe that inserting/extracting oxygen into/from to LaSrMnO₃, similarly to Mefford's approach, will provide a beautiful tool to reversibly control magnetism. Indeed it is known that oxygen plays an important role in mediating the magnetic exchange interactions in manganites (see for example Coey et al., *Adv. Phys.* 58 (2009)). In our group we have already performed some experiments of ionic intercalation (with Li⁺) into bulk ferromagnets (for example see Dasgupta et al., *Adv. Mater.* 26 (2014) or Dasgupta et al., *Adv. Funct. Mater.* 26 (2016)). In the future we would like to expand the idea to oxygen intercalation to many other systems, including LaSrMnO₃. In this sense, as the reviewer pointed out, the work of Dr. T. Lim and Dr. J. Gracia will be helpful for the understanding and interpretation of the role of oxygen in relation to the magneto-electric coupling.

It is not clear why the authors chose such a specific ionic liquid vs other media. Their electrolyte is somewhat exotic and as perhaps others have commented hard to make water and oxygen free which could bias some of their measurements. This reviewer is curious as to if such magnetic effects can be evidenced in strong basic electrolytes relevant to water oxidation and anion based pseudocapacitance. Perhaps theory could assist in that this choice of ionic liquid creates a very special field assisted ordering of the double layer of ions which also influences magnetic ordering. To my knowledge there are no examples of such experiments or calculations to support this concept. However, perhaps these results provide impetus for new paradigms for understanding such complex behaviors in more complex interfacial processes. Most experiment and theory is stuck on single metal solid/liquid interfaces and have yet to evolve to well controlled metal oxide electrochemical interfaces.

As such this paper stands on its own merit and should be published, yet much work is ahead to understand complicated chemical and physical mechanisms of complex metal oxides.

The choice of using the DEME/TFSI ionic liquid stems from its good electrochemical (potential window of about $\Delta V \pm 3V$) and temperature (decomposition at 650 K) stability. Other groups already used this kind of ionic liquid (for example, Yuan et al., *Adv. Funct. Mater.* 19 (2009); Zhou et al., *J. Appl. Phys.* 111 (2012) and Cui et al., *Adv. Funct. Mater.* 24 (2014)). We took care of purifying the ionic liquid from water contamination by keeping it under vacuum conditions (10^{-3} mbar) at 120 °C for 24 hours; Karl-Fisher titration revealed an H₂O level < 50 ppm. The preparation of the electrochemical tuning cell prior to the magnetic tuning experiments was performed under glove box conditions. Little amounts of water contamination might occur during the transfer from the glovebox to the SQUID magnetometer (only a few minutes). Once in the SQUID magnetometer the tuning cell was in a dry (a few mbar of He) atmosphere.

We agree with the reviewer that it would be interesting to test other kinds of aqueous and non-aqueous electrolytes. A few years ago our group performed some experiments on LSMO powders using an aqueous electrolyte (please, see Mishra et al., *J. Appl. Phys.* 113 (2013)), but the magnetic tuning effect was much smaller (around 2 %) comparing to the present study (33 %). At that time we did not

investigate the possibility of intercalation pseudocapacitance. As previously stated, it will be interesting to test oxygen intercalation in LSMO (a good starting point will be probably to use the same electrolytes reported in the work of Mefford).

We totally agree with the reviewer that theory can and will assist the experimental research in the interpretation and modelling of the interfacial charging/discharging processes as well as the evolution of magnetic phase separation upon application of external voltages. We do hope that our work will trigger the interest of the theoretical communities working on pseudocapacitors and magnetoelectrics to find some common ground for fruitful and stimulating discussion.